# Functional connectivity and GABAergic signaling modulate the enhancement effect of neurostimulation on mathematical learning

George Zacharopoulos[1,2]*, Masoumeh Dehghani[3,4], Beatrix Krause-Sorio[5], Jamie Near[6,7], Roi Cohen Kadosh[1,8]*

**1** Wellcome Centre for Integrative Neuroimaging, Department of Experimental Psychology, University of Oxford, Oxford, United Kingdom, **2** School of Psychology, Swansea University, Swansea, United Kingdom, **3** Department of Psychiatry, McGill University, Montreal, Canada, **4** Centre d'Imagerie Cérébrale, Douglas Mental Health University Institute, Montreal, Canada, **5** The Richard M. Lucas Center for Imaging, Stanford University, Stanford, California, United States of America, **6** Physical Sciences, Sunnybrook Research Institute, Toronto, Canada, **7** Department of Medical Biophysics, University of Toronto, Toronto, Canada, **8** School of Psychology, University of Surrey, Guildford, United Kingdom

\* g.zacharopoulos@swansea.ac.uk (GZ); r.cohenkadosh@surrey.ac.uk (RCK)

## Abstract

Effortful learning and practice are integral to academic attainment in areas like reading, language, and mathematics, shaping future career prospects, socioeconomic status, and health outcomes. However, academic learning outcomes often exhibit disparities, with initial cognitive advantages leading to further advantages (the Matthew effect). One of the areas in which learners frequently exhibit difficulties is mathematical learning. Neurobiological research has underscored the involvement of the dorsolateral prefrontal cortex (dlPFC), the posterior parietal cortex (PPC), and the hippocampus in mathematical learning. However, their causal contributions remain unclear. Moreover, recent findings have highlighted the potential role of excitation/ inhibition (E/I) balance in neuroplasticity and learning. To deepen our understanding of the mechanisms driving mathematical learning, we employed a novel approach integrating double-blind excitatory neurostimulation—high-frequency transcranial random noise stimulation (tRNS)—and examined its effect at the behavioral, functional, and neurochemical levels. During a 5-day mathematical learning paradigm ($n = 72$) active tRNS was applied over the dlPFC or the PPC, and we compared the effects versus sham tRNS. Individuals exhibiting stronger positive baseline frontoparietal connectivity demonstrated greater improvement in calculation learning. Subsequently, utilizing tRNS to modulate frontoparietal connectivity, we found that participants with weaker positive baseline frontoparietal connectivity, typically associated with poorer learning performance, experienced enhanced learning outcomes following dlPFC-tRNS only. Further analyses revealed that dlPFC-tRNS improved learning outcomes for participants who showed reductions in dlPFC GABA when it was accompanied by a reduced positive frontoparietal connectivity, but this effect was

**Data availability statement:** All relevant data are within the manuscript and its Supporting information files.

**Funding:** Funding: This research was supported by the European Research Council (Learning&Achievement 338065 to RCK, https://erc.europa.eu/) and the Wellcome Trust (0883781 to RCK, https://wellcome.org/). The funders had no role in study design, data collection and analysis, decision to publish, or preparation of the manuscript.

**Competing interests:** I have read the journal's policy and the author RCK of this manuscript has the following competing interests: RCK serves on the scientific advisory boards of Neuroelectrics Inc., Innosphere Inc., is the founder and shareholder of Cognite Neurotechnology Ltd., and is an Editorial Board Member at PLOS Biology.

**Abbreviations:** dlPFC, dorsolateral prefrontal cortex; E/I, excitation/inhibition; MPRAGE, magnetization prepared rapid acquisition with gradient echo; PFC, prefrontal cortex; PPC, posterior parietal cortex; SPECIAL, spin-echo full-intensity acquired localized; tRNS, transcranial random noise stimulation; VAPOR, variable power radio frequency pulses with optimized relaxation delays; VOI, voxel of interest; VTOC, ventral temporal occipital cortex; WIAT, Wechsler Individual Achievement Test; WIN, Wellcome centre for human neuroimaging.

reversed for participants who showed increased positive frontoparietal connectivity. Our multimodal approach elucidates the causal role of the dlPFC and frontoparietal network in a critical academic learning skill, shedding light on the interplay between functional connectivity and GABAergic modulation in the efficacy of brain-based interventions to augment learning outcomes, particularly benefiting individuals who would learn less optimally based on their neurobiological profile.

## Introduction

Academic learning has profound implications for individuals and society at large [1–3]. However, not all individuals benefit equally from educational opportunities, a phenomenon referred to as the Matthew effect [4] in education. This cognitive disparity is a significant societal issue, exacerbating inequalities in learning and perpetuating gaps in education, which, in turn, limit access to future resources and opportunities. Mathematical learning, in particular, presents a considerable barrier for many. For instance, an organisation for economic cooperation and development (OECD) report from 2016 revealed that 24%−29% of adults in developed countries such as the United States, the United Kingdom, Germany, and France possess mathematical skills at or below the level expected of 5–7-year-olds [5]. This widespread deficiency in mathematical skills has extensive consequences, including higher unemployment, reduced economic growth, poorer health outcomes, diminished political engagement, and lower levels of trust in others [5]. In our increasingly science, technology, engineering, mathematics (STEM)-oriented societies, the repercussions of low mathematical competence are likely to intensify [6].

Longitudinal studies suggest that mathematical abilities are relatively stable from childhood through adulthood, driven primarily by biological rather than environmental factors [7–9]. Genetic research has identified correlations between mathematical abilities and genes associated with potassium channels, which are essential for neuronal excitability. Variations in these genes are linked to learning disabilities and neurodevelopmental delays, underscoring their role in neuronal excitability and synaptic plasticity [10–12]. However, despite these insights, the causal mechanisms underpinning mathematical learning remain poorly understood, with much of the existing research focusing on neural correlates rather than causal factors [13–16].

Non-invasive neurostimulation techniques offer a promising avenue for exploring the causal mechanisms of mathematical learning by modulating cortical excitability and neurochemicals involved in cognitive processing [17–19]. To advance our understanding of the neurobiological underpinnings of mathematical learning, we employed a double-blind excitatory neurostimulation protocol—high-frequency (100–640 Hz) transcranial random noise stimulation (tRNS)—in conjunction with proton Magnetic Resonance Spectroscopy ($^1$H-magnetic resonance spectroscopy (MRS)) and resting-state functional magnetic resonance imaging (fMRI) over a 5-day mathematical learning paradigm involving 72 healthy young adults. The dorsolateral prefrontal cortex (dlPFC) and the posterior parietal cortex (PPC) were selected as voxels of interest (VOIs) based on their established roles in skill acquisition and mathematical proficiency [20–27].

Neurocognitive models have suggested that arithmetic information is processed in a multimodal and distributed manner in the human brain. Accounts based on Bayesian meta-analytic models identified the canonical circuit of arithmetic problem solving [28], which consists of brain regions that are mostly localized to the dorsal aspects of the PPC, ventral temporal-occipital cortex (VTOC), and the prefrontal cortex (PFC), including the premotor cortex and the inferior and middle frontal gyri [29].

Previous accounts for how arithmetic information is mastered indicated that the development of numerical and arithmetic abilities is enabled by a significant increase in working memory [30]. Indeed, the role of working memory and executive functions is a crucial feature in several theoretical models of cognitive learning, in general, and arithmetic learning in particular (A Triarchic Theory of Learning, [20]). These models predict that mathematical procedures (calculation), which rely heavily on executive functions including working memory, attention processes, inhibition, and shifting [31] are typically associated with the dlPFC [32]. This view, which is supported by previous work on the acquisition of arithmetic expertise in normally developing children and adults, revealed that a positive shift in arithmetic competence corresponds to a shift from effortful processing that engages the prefrontal regions, such as the dlPFC, to skilled and fast retrieval that involves parietal regions such as the PPC [21]. This is because repeating the arithmetic problems strengthens a direct retrieval of the correct solution from memory (e.g., $23 \times 8 = 184$). Concurrently, the demand on quantity-based processing and complex processes, such as working memory (e.g., $23 \times 8 = [20 \times 8] + [3 \times 8] = 160 + 24 = 184$), monitoring and attention, sustained by frontal areas, is diminished [21]. Accordingly, we hypothesize that tRNS over the dlPFC, while participants learn to apply a new algorithm, will improve calculation learning. Based on a previous tRNS study using the same paradigm as in this work we predict that the effect will be found for reaction times (RTs), rather than accuracy [33]. In contrast, we do not expect to find a reliable tRNS (over the dlPFC) effect for drill learning, which does not require the application of mathematical procedures. However, tRNS over the PPC may improve drill learning, but not calculation learning. While in a previous study no effect was found when tRNS was applied over the PPC during calculation and drill learning [33], the same parietal montage and tRNS parameters as in our study [33] improved performance in a vocabulary learning task, which involves some component of drill learning [34].

Apart from the aforementioned dissociated roles of the frontal and the parietal regions, these regions, which are co-activated during arithmetic tasks [29], form the frontoparietal networks that support the manipulation of numerical information, which is crucial for calculation learning. Apart from frontoparietal networks, the frontohippocampal networks are involved in creating associative memories and connecting new information with existing knowledge, which contributes to the development of long-term memories that enable generalization beyond specific problem characteristics [35]. Taking these results together with previous work showing that cortico-hippocampal connectivity is involved in mathematical learning [36], we hypothesize that frontoparietal connectivity is associated mainly with calculation learning, while frontohippocampal connectivity is associated with both drill and calculation learning performance.

Beyond the brain regions and networks involved in mathematical learning, previous studies, have highlighted the crucial role of excitatory and inhibitory neural circuits in shaping neural excitation/inhibition (E/I) balance. E/I balance governs the onset and closure of sensitive periods in brain development and skill acquisition [37–39]. For example, in humans, research on visual procedural learning has utilized glutamate and GABA measurements to quantify E/I balance, associating high E/I with plasticity (the capacity for change in learning mechanisms) and lower E/I with stability (the resistance to change in these mechanisms) [40]. Effective learning requires an initial phase of plasticity, followed by a phase of stability to consolidate knowledge and prevent interference from subsequent learning experiences [40]. However, the role of plasticity and stability in more complex learning domains, such as mathematics, remains underexplored [41]. In this study, we employed tRNS due to its potential to change the levels of excitation and inhibition [41–44]. Similar to previous neuroimaging studies we noninvasively quantified excitation and inhibition using glutamate and GABA, respectively [45–47]. tRNS, due to its potential excitatory effect, has been suggested to enhance plasticity with a continuous effect beyond the stimulation session, as indicated by lasting effects beyond the stimulation period (e.g., weeks to months; [33,34,42,48–52]. Our approach

is further motivated by the association between individual variability in glutamate and GABA levels and mathematical attainment [15]. Therefore, we hypothesize that changes in glutamate and GABA levels will predict calculation and drill learning.

Participants engaged in a 5-day mathematical learning task, solving new arithmetic problems through either calculation (algorithm-based) or drill (rote memorization) [53,54] (Fig 1). The calculation method required the integration of declarative, procedural, and conceptual knowledge, all of which are critical for mathematical learning [55]. To provide a comprehensive network-level understanding of frontoparietal stimulation, we utilized resting-state fMRI to assess the functional connectivity between our predefined regions throughout the learning process.

We hypothesize that participants who exhibit a greater increase in inhibitory neurochemical levels over the 5 days will benefit more from tRNS, as this indicates a more rapid transition from plasticity to stability, thereby facilitating better learning outcomes. Additionally, given the established link between neurochemical concentrations and functional connectivity [16,45,56], we explored the interplay between neurochemicals (GABA and glutamate) and functional connectivity to elucidate the complex mechanisms underlying mathematical learning.

## Materials and methods

### Participants

We recruited 72 healthy, right-handed participants (36 males, mean age = 21.75, SD = 3.21, range = 18–30), and 36 females (mean age = 22.1, SD = 2.63, range = 19–28). Participants were randomly assigned into three stimulation groups (i.e., sham tRNS versus active dlPFC-tRNS versus active PPC-tRNS), and each of the three stimulation groups consisted of 12 females and 12 males. All the participants provided written consent before the experiment started. The study was approved by the Berkshire Research Ethics Committee (National Research Ethics Service Committee South Central—Berkshire, approval number: 10/H0505/72) and was conducted according to the principles expressed in the Declaration of Helsinki.

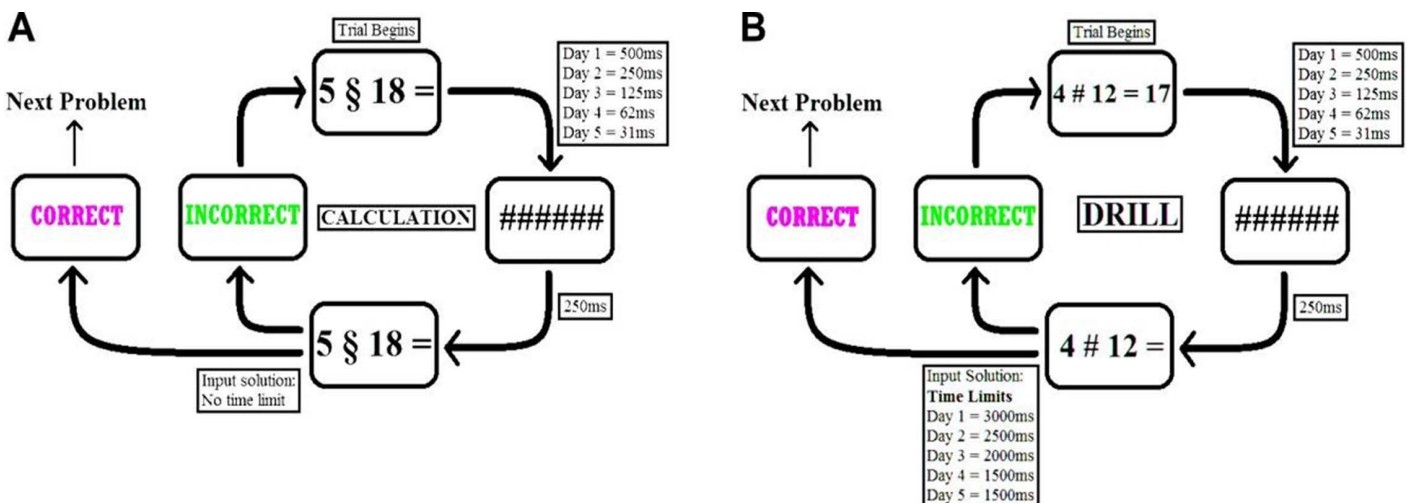

**Fig 1. Task procedure.** Calculation and drill problems were presented to the participants in different blocks on each learning day (see Materials and methods). **(A)** Calculation problems (denoted by "§") began with the presentation of a problem, a brief mask, and an unlimited input duration. Upon giving the correct response, the participant was presented with a new problem. Incorrect responses led to the repetition of the same problem until the correct response was provided. **(B)** Drill problems (denoted by "#") were presented as a combination of two operands and an outcome. In contrast to the calculation problems, in the drill problems, the underlying algorithm to derive the result by calculation was not known. The combination had to be memorized and reproduced after masking. Both presentation time and response input time decreased from one session to the next, overall. Problems were repeated when answered incorrectly.

### Learning task

In the current study, we used mathematical learning as a model for studying academic learning. The mathematical learning paradigm used here was adopted from Delazer and colleagues [53]. Participants learned to solve equations either by memorizing solutions to given problems ("drill learning," denoted by "#") or by using an algorithm ("calculation learning," denoted by "§") (Fig 1 and S1 Text).

### tRNS

tRNS uses a rapidly alternating current at a fixed range of frequencies to induce cortical excitation [27] and provides adequate participant-blinding in sham-controlled research [57,58]; the current was applied via electrodes on the scalp (see below for details). Electrodes were placed above the bilateral dlPFC (F3 and F4) or PPC (P3 and P4), as defined by the international 10–20 system for EEG recording (for additional information, see S1 Text).

### MRS

Single VOI $^1$H-MRS data were acquired using a 3T Verio system (Siemens Healthcare, Erlangen, Germany) at the Wellcome Centre for Integrative Neuroimaging (WIN) at the University of Oxford. The system body coil was used for signal transmission and a 32-channel head receive array (HEA-HEP; Siemens Healthcare, Erlangen, Germany) was used for signal reception. T1-weighted MR images with a slice thickness of 1 mm were acquired (MPRAGE; magnetization prepared rapid gradient echo) with repetition time (TR) = 2,040 ms, echo time (TE) = 4.68 ms, TI = 900 ms (inversion time), and a flip angle of 8°. The localized MRS sequence SPECIAL (spin-echo, full-intensity, acquired, localized) technique was used to acquire 128 averages with a TR = 4,000 ms, a TE = 8.5 ms, a bandwidth of 4,000 Hz, and 4,096 points [59,60]. Water suppression was performed using VAPOR (variable power radio frequency pulses with optimized relaxation delays) [61]. Three 2 × 2 × 2 cm voxels of interest were manually localized on axial and coronal slices and placed over the left PPC (S1 FigA) or the left dlPFC (S1 FigB), depending on the tRNS condition, and a control region in the primary visual cortex (V1) of each participant. For additional information, including MRS data analyses, see S1 Text.

### Resting fMRI

Functional images were acquired with an interleaved sequence (TR = 2410 ms, TE = 30 ms, flip angle 90°, number of slices = 44, voxel dimensions = 3 × 3 × 3 mm, number of volumes = 128). fMRI data were preprocessed and analyzed using the CONN toolbox (www.nitrc.org/projects/conn, RRID: SCR_009550) in SPM12 (Wellcome Department of Imaging Neuroscience, Institute of Neurology, London, UK) using a standard preprocessing pipeline that involves MNI-space normalization [62]. For additional information, including the calculation of the dlPFC-PPC (henceforth, frontoparietal) functional connectivity, see S1 Text.

### Procedure

Participants were screened for safety over the phone and invited to the WIN at the University of Oxford (for the scanning sessions). A scan (i.e., structural MRI, $^1$H-MRS, and resting-state fMRI) was performed, and each participant was subsequently invited to the Department of Experimental Psychology for 5 days of tRNS and mathematical learning (for the behavioral sessions). On the first day before the learning sessions started, mathematical attainment was assessed using the Wechsler Individual Achievement Test (WIAT) [63], as has been done previously [64]. This standardized test (Mean score = 100, SD = 15) includes Numerical Operations, which assesses written calculation abilities, and Mathematical Reasoning, which assesses verbal word problem mathematical abilities. We used the composite score (henceforth, the mathematical attainment) to match the participants in each group according to their mathematical abilities. Computerized drill learning and calculation learning, paired with active tRNS condition or sham tRNS condition, were provided for 30 min.

After the last learning session, participants underwent another scanning session involving the same sequence of events, including repeated safety screening.

## Statistical analyses

We performed linear, mixed-effects analyses with learning as the dependent variable (mean RT of correct trials only) and using several time points (i.e., day [5 as there were 5 days] and learning type [2 levels: drill and calculation]). For additional information, see S1 Text.

## Results

### Baseline frontoparietal positive connectivity predicted improved learning

First, we examined whether mathematical (i.e., "calculation" or "drill") learning was influenced by the baseline (pre-tRNS) frontoparietal functional connectivity (Fig 2A). To control for the confounding effect of tRNS, we focused on the sham tRNS

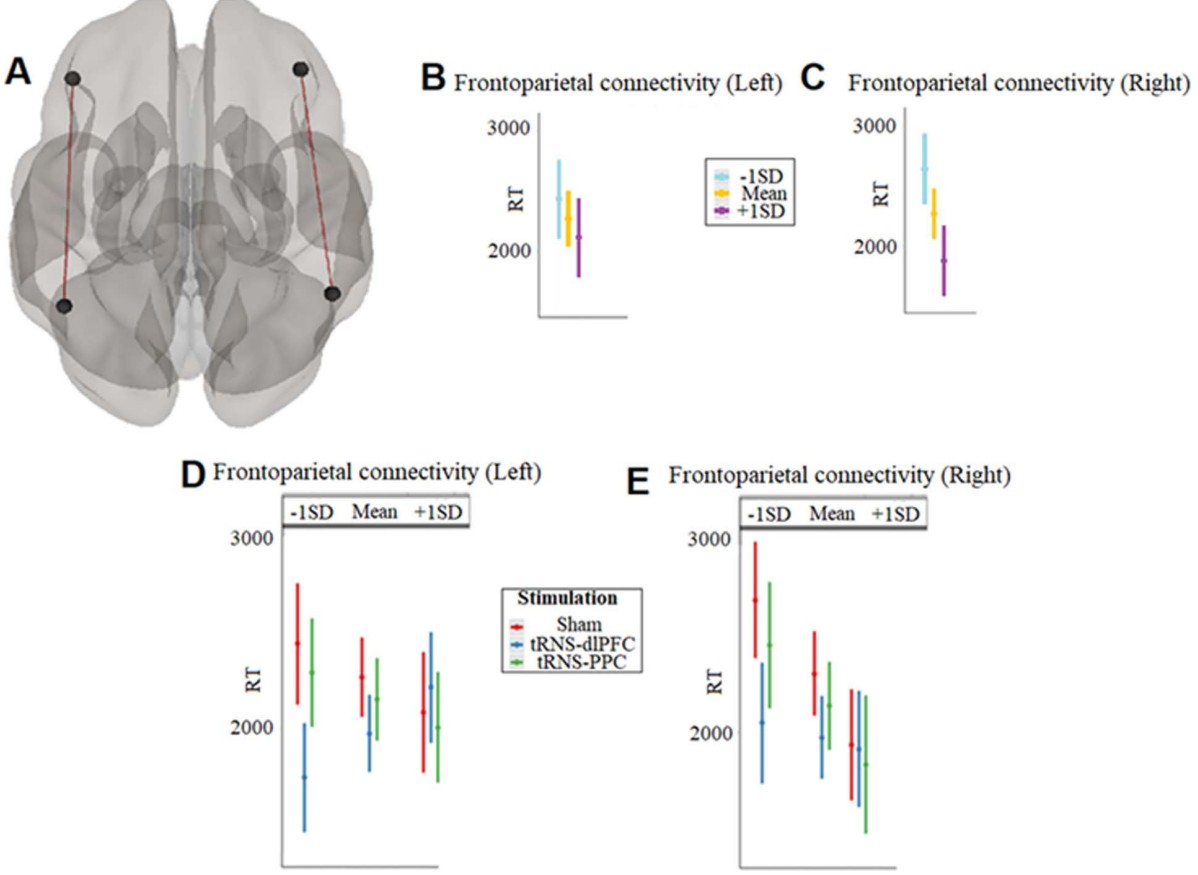

Fig 2. (A) A 3D volume, generated manually in CONN [62], depicting the four frontoparietal seeds (left dlPFC, right dlPFC, left PPC, right PPC) as well as the right and left frontoparietal connectivity that was used in the functional connectivity analyses. Predicting RT in the calculation learning task (for the corresponding analyses depicting the drill learning task, see S3 Fig) using the left and right frontoparietal connectivity (Panels B and C, respectively). Predicting RT in the calculation learning task using the left and right frontoparietal connectivity (Panels D and E, respectively) as a function of tRNS condition. Following Aiken and colleagues' [65] suggestion for plotting continuous variables, we plotted the results here and in the other figures for one SD above the mean (i.e., mean + 1 SD), the mean, and one SD below the mean (i.e., mean − 1 SD). Error bars correspond to the 95% standard error of the mean. The data underlying the results in the Panels (B–E) can be found in S2 Data. dlPFC, dorsolateral prefrontal cortex; PPC, posterior parietal cortex.

condition only. We predicted learning using the variables day, learning type (drill versus calculation), and baseline frontoparietal connectivity. We observed an interaction between frontoparietal connectivity*learning type both for the left (Fig 2B, $B=2,462$, SE$=839$, $T_{210}=2.9$, $P=0.004$, CI = [808, 4,116]) and for the right connectivity (Fig 2C, $B=1,960$, SE$=619.0$, $T_{210}=3.17$, $P=0.002$, CI = [740, 3,180]). In both cases, the effect was restricted to the calculation learning; individuals with stronger baseline positive connectivity showed a better performance.

Based on previous work showing that cortico-hippocampal connectivity is involved in mathematical learning [36], we assessed in a separate analysis whether dlPFC- or PPC-hippocampus connectivity explains variance in calculation or drill learning (see S1 Table). We found two significant effects: the left dlPFC–hippocampus connectivity ($B=2,650$, SE$=988$, $T_{22}=2.68$, $P=0.01$, CI = [601, 4,699]) and the left dlPFC–hippocampus connectivity*learning type interaction ($B=-2,720$, SE$=1,045$, $T_{210}=-2.6$, $P=0.01$, CI=[-4,780, -661]), where more positive connectivity was associated with poorer calculation learning, again with no significant differences for drill learning (S2 Fig).

We subsequently assessed whether left dlPFC–hippocampus connectivity would explain the additional variance in calculation learning above and beyond left frontoparietal connectivity. However, the left dlPFC–hippocampus baseline connectivity was no longer significant, whereas the frontoparietal baseline connectivity predictors were still significant (S2 Table).

### Active tRNS over the dlPFC improved learning

Second, we examined whether tRNS induced changes in mathematical learning. As can be seen in Table 1, active tRNS over the dlPFC improved calculation learning.

### Active tRNS over the dlPFC in individuals with weaker positive baseline frontoparietal connectivity improved learning

To examine whether the frontoparietal connectivity plays a causal role in mathematical learning we modulated its activity by using active tRNS. By targeting one node (dlPFC) or another (PPC) in different individuals, we tested whether we could enhance learning. We predicted learning using the tRNS condition (sham tRNS versus dlPFC-tRNS versus PPC-tRNS), day, (1–5), learning type (drill versus calculation), and baseline frontoparietal connectivity (for additional

**Table 1. Behavioral results when predicting RT from stimulation condition (compared to sham tRNS, which serves as the reference), day, and learning type. Statistics: Value=regression coefficient, SE=standard error, DF=degrees of freedom, T=t-value, p=p-value, dlPFC=dorsolateral prefrontal cortex, PPC=posterior parietal cortex, tRNS, transcranial random noise stimulation. Interactions are denoted by the * symbol. The data underlying the results in this table can be found in S1 Data.**

| | Value | SE | DF | T | P | CI_L | CI_U |
|---|---|---|---|---|---|---|---|
| (Intercept) | 3,152.746 | 150.463 | 639 | 20.954 | <0.0001 | 2,857.283 | 3,448.209 |
| TypeDrill | −2,373.140 | 166.636 | 639 | −14.241 | <0.0001 | −2,700.360 | −2,045.920 |
| Day | −297.111 | 35.527 | 639 | −8.363 | <0.0001 | −366.875 | −227.347 |
| dlPFC-tRNS | −535.643 | 196.507 | 69 | −2.726 | 0.0081 | −927.664 | −143.622 |
| PPC-tRNS | −343.411 | 209.225 | 69 | −1.641 | 0.1053 | −760.804 | 73.983 |
| TypeDrill*Day | 252.765 | 50.243 | 639 | 5.031 | <0.0001 | 154.104 | 351.425 |
| TypeDrill*dlPFC-tRNS | 543.069 | 205.446 | 639 | 2.643 | 0.0084 | 139.638 | 946.499 |
| TypeDrill*dlPFC-PPC | 260.252 | 229.191 | 639 | 1.136 | 0.2566 | −189.807 | 710.310 |
| Day*dlPFC-tRNS | 82.866 | 43.801 | 639 | 1.892 | 0.0590 | −3.146 | 168.878 |
| Day*PPC-tRNS | 51.874 | 48.864 | 639 | 1.062 | 0.2888 | −44.079 | 147.827 |
| TypeDrill*Day*dlPFC-tRNS | −94.401 | 61.944 | 639 | −1.524 | 0.1280 | −216.040 | 27.238 |
| TypeDrill*Day*PPC-tRNS | −45.750 | 69.104 | 639 | −0.662 | 0.5082 | −181.448 | 89.947 |

information, see S1 Text). We observed a tRNS condition*frontoparietal connectivity*learning type interaction for the left (Fig 2D, $B = -3,312$, SE = 1,007, $T_{603} = -3.3$, $P = 0.001$, CI = [$-5,289$, $-1,335$]) and the right hemisphere (Fig 2E, $B = -2,291$, SE = 838, $T_{603} = -2.7$, $P = 0.006$, CI = [$-3,937$, $-646$]). In both cases, we observed that the effect was restricted to calculation learning, showing a difference only between the sham and dlPFC-tRNS conditions. Participants who were predicted to perform poorly due to weaker positive baseline frontoparietal connectivity, as observed in their peers in the sham tRNS condition, performed as well as or even better than those who were expected to outperform them when they received dlPFC-tRNS. Those who received PPC-tRNS showed a similar pattern to those in the sham tRNS condition (Fig 2D–2E). These findings were specific to frontoparietal connectivity, indicating that the effect of dlPFC-tRNS on learning depends on the baseline levels of frontoparietal connectivity, but not occipitofrontal or occipitoparietal functional connectivity (for statistical details, see S3 Table).

To assess whether the effect of active tRNS is confounded by baseline mathematical ability we performed an additional analysis with the inclusion of three covariates (i.e., mathematical attainment, first block drill RT, and first block calculation RT). The effect of active tRNS was not confounded by baseline mathematical ability (S4 Table).

In contrast to our predictions, active tRNS did not reliably alter GABA or glutamate levels (S5 Table). We examine their moderating role on the effect of dlPFC-tRNS on learning as indicated by the interaction between Δ dlPFC neurochemical level (i.e., post-tRNS – baseline neurochemical concentrations) and Δ frontoparietal functional connectivity (Materials and methods, S1 text, and S6 Table). We observed a tRNS condition*Δ dlPFC GABA*Δ right frontoparietal connectivity*learning type interaction (Fig 3, $B = 18,110$, SE = 6,651, $T_{120} = 2.7$, $P = 0.007$, CI = [4,942, 31,278]). This interaction remained significant after controlling for Δ dlPFC glutamate ($B = 29,571$, SE = 8,523, $T_{111} = 3.47$, $P = 0.0007$, CI = [12,682, 46,460]). Compared to the sham tRNS, dlPFC-tRNS induced the largest improvement in calculation learning for participants who showed reductions in dlPFC GABA and positive frontoparietal connectivity ($P = 0.0148$, Fig 3A). However, if an increased, rather than decreased, frontoparietal connectivity was observed, the performance was detrimental ($P < 0.0001$, Fig 3A).

A closer examination of the sham tRNS group (red color in Fig 3) suggests that greater positive connectivity increase was beneficial for those in the plasticity state as indicated by reduced GABA (−1SD change in GABA, $P = 0.0001$, Fig 3A and S7 Table A) but was detrimental for those in the stability state (+1SD change in GABA, $P = .013$, Fig 3C and S7 Table A).

In contrast, for the active tRNS group (blue color in Fig 3), compared to those with a smaller positive connectivity increase, a greater positive connectivity increase was detrimental both in the plasticity ($P = 0.0014$, Fig 3A and S7 Table B) and the stability state ($P = 0.009$, Fig 3C and S7 Table B).

## Discussion

In this study, we examined how tRNS can alter mathematical learning and the neural mechanisms (i.e., GABA, glutamate, and functional connectivity) it may alter to allow optimize learning. Three main findings emerged: (1) baseline positive frontoparietal connectivity predicted calculation learning, where stronger positive connectivity was associated with better learning; (2) dlPFC-tRNS improved calculation learning in individuals with a baseline connectivity profile that would predict otherwise poor learning; and (3) dlPFC-tRNS improved calculation learning for participants who showed reductions in dlPFC GABA when it was accompanied by a reduced positive frontoparietal connectivity, but this effect was reversed for participants who showed increased positive frontoparietal connectivity.

### The role of baseline frontoparietal connectivity in predicting learning and tRNS efficacy

Our connectivity analyses revealed that baseline frontoparietal connectivity interacts with learning type in predicting learning: individuals with a stronger positive baseline frontoparietal connectivity showed better learning that was specific to calculation.

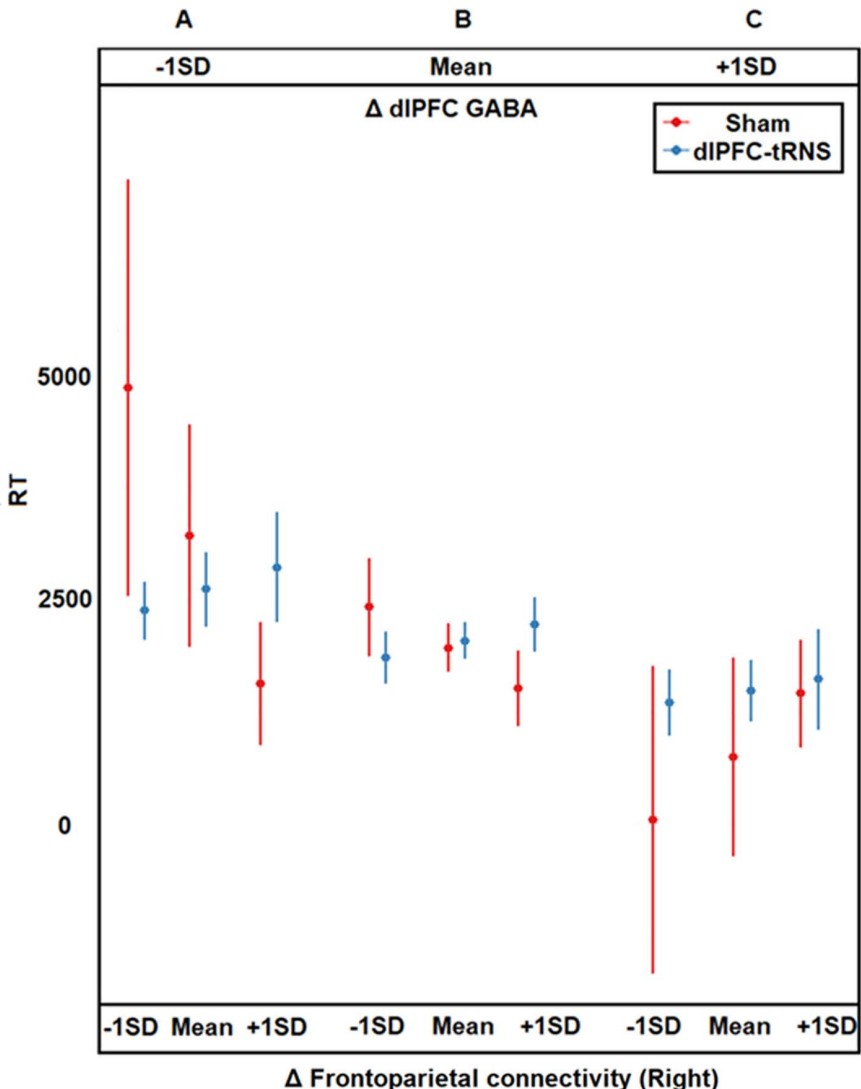

**Fig 3. Plotting the simple three-way interaction of tRNS\*right frontoparietal connectivity\*dlPFC GABA.** This is based on the four-way interaction between learning type\*tRNS condition\*right frontoparietal connectivity across three levels of the dlPFC GABA. For simplicity, we display the calculation learning results. For the corresponding drill learning results, see S4 Fig. Participants who showed reductions in positive frontoparietal connectivity and reductions in dlPFC GABA showed calculation learning improvement in the dlPFC-tRNS (vs. sham tRNS) condition. The effect of the dlPFC-tRNS (vs. sham tRNS) condition is reversed for participants who showed increases in positive frontoparietal connectivity and reductions in dlPFC GABA. dlPFC-tRNS benefited those participants who performed the worst, namely, the participants in the sham tRNS condition who showed the most pronounced reductions in positive frontoparietal connectivity and the most pronounced reductions in dlPFC GABA (Fig 3A, leftmost subpanel). Following Aiken and colleagues' [65] suggestion for plotting continuous variables, we plotted the results for one SD above the mean (i.e., mean + 1 SD), the mean, and one SD below the mean (i.e., mean − 1 SD). Error bars correspond to the 95% standard error of the mean. The data underlying Fig 3 can be found in S3 Data. dlPFC, dorsolateral prefrontal cortex; tRNS, transcranial random noise stimulation.

Furthermore, individuals with weaker positive baseline connectivity, who would have otherwise shown poorer calculation learning, improved their calculation learning when they received active tRNS over the dlPFC compared to the sham tRNS condition. Again, we obtained this effect specifically for calculation learning. By contrast, stimulating the other node in this network, the PPC did not yield a similar effect. Taken together, these findings suggest that the key causal effect of connectivity on learning is rooted in the dlPFC. The enhancement effect of those with predicted poorer calculation learning

raises the possibility that tRNS has a "scaling up" effect that can release the brakes due to suboptimal brain activity to support learning [66,41]. The fact that individuals with stronger positive baseline connectivity showed better calculation learning suggests a neurobiological equivalent of the Matthew effect. The Matthew effect [4] has been shown in different types of learning, including mathematical learning [67], language learning [68,69], and memory-related processes such as vocabulary growth [69], whereby a baseline cognitive gap between high and low achievers leads to an even greater increase in inequality between these two groups.

Our results, in line with previous results [7–9], support the view that such differences may stem from biological origin. However, such differences may be independently influenced by socioeconomic status and polygenetic factors [70]. By using a learning task and modulating the neural corelates involved in mathematical learning, we were able to show that the Matthew effect can be explained and mitigated at the biological level. Our results do not undermine the environmental factors (e.g., socioeconomic status) that could also draw a causal effect but are less likely to play a role in this study due to the random allocation of participants and the use of a learning task. Our results highlight the possibility of reversing such biological disadvantage using an excitatory form of neurostimulation to target the neural substrates involved in calculation learning. This is in line with a recent study that identified genes that correlate with mathematical abilities, as crucial for neuronal excitability [9]. Future studies can build on our findings by assessing how long this effect lasts and whether such an approach can have practical relevance outside the controlled lab environment. Answering these questions would have important implications in the fields of education and the science of learning.

Furthermore, the application of active tRNS, our excitatory neurostimulation protocol, sheds light on the involvement of the frontoparietal network in skill acquisition in calculation learning but not by drill learning. Unlike in calculation learning, in drill learning, active tRNS did not produce a robust enhancement effect. These behavioral results are in contrast to a previous study [33], but in line with another [71]. It might indicate that the effect of tRNS on mathematical drill learning is less robust. This might be attributed to the ease of acquisition of such learning, which required shallower processing during the encoding stage, and therefore faster learning [72]. This, in turn, may lead to an early shift from plasticity to stabilization [40], and therefore reduce the beneficial excitability effect of tRNS.

### Active tRNS effect is mitigated by changes in dlPFC GABA

Subsequently, we revealed a multimodal interaction whereby, compared to the sham tRNS, the dlPFC-tRNS induced an improvement in mathematical learning for participants who showed reduced dlPFC GABA and reduced positive frontoparietal connectivity. This beneficial effect was reversed if the participants showed increased positive frontoparietal connectivity. In the sham condition, the worst-performing individuals were those who exhibited the most pronounced reductions in positive frontoparietal connectivity and dlPFC GABA (Fig 3A, leftmost subpanel). However, dlPFC-tRNS benefited the participants with the same neurobiological profile.

How do these results fit with the putative mechanisms of tRNS [26,66,73]? Different studies have attributed the effect of tRNS to angiogenesis [33], stochastic resonance [26,66,71], E/I [41], a reduction in GABA precursor [43], and sodium channels [44]. It would be simplistic to assume that a single mechanism can explain the effect of tRNS. Rather, factors such as the timescale in which tRNS was applied, the stimulated region, the cognitive domain, the level of skill acquisition, and the neural state during stimulation, which vary across studies, may play an important role in the effect of tRNS and the underlying mechanisms on learning [74]. The beneficial effect of tRNS on those who would otherwise underperform is in line with the principle of stochastic resonance [26,66,71], a phenomenon by which the addition of random noise to a weak signal enhances the detectability of that signal [75]. Previous studies have demonstrated performance enhancements both when the random noise was added directly to the weak perceptual stimuli or via tRNS over cortical regions [27,52,66,76]. The current results suggest that the tRNS effect may have been achieved by shifting subthreshold receptor potentials within the dlPFC toward the firing threshold, increasing the likelihood of action potentials in neurons that would otherwise not reach the threshold [66]. This, in turn, is likely to increase the E/I level, as indicated by decreased GABA

concentration. However, the null result in our MRS measurement create challenge to this explanation. The reason for this null result could be multiple, including the length of our training that span over 5 days and might have led to a shift from plasticity to stability, resulting in a spurious E/I effect [40], and the hours that passed from the end of the learning and tRNS to the scanning session, which can show fluctuations in E/I dynamics [77].

## The importance of stability and plasticity states during mathematical learning and the interplay between functional connectivity and GABAergic modulation in the efficacy of brain-based interventions to augment learning outcomes

Previous studies on visual procedural learning demonstrated that neural E/I regulates the transition between stability and plasticity states (for a review, see [40]). Here we extend these previous studies by demonstrating how the states of plasticity and stability, as indicated by the relative change in dlPFC GABA concentration, impact mathematical learning. Initially, we showed that overall performance (i.e., across stimulation conditions and connectivity levels) is better in those in the stability state (+1SD GABA, Fig 3C) than those in the plasticity stage (−1SD GABA, Fig 3A). This set of findings suggests that more stability is associated with better calculation learning.

Importantly, however, by employing a multimodal experiment, we showed that the mechanisms that regulate learning improvement are more complex than described before, and co-depend on both functional connectivity and GABA concentration. Normally, as exhibited by the sham tRNS group, a greater increase in positive connectivity is beneficial for those individuals in the plasticity state (−1SD change in GABA, Fig 3A) and for those neither in the plasticity nor stability state (Mean change in GABA, Fig 3B) but detrimental for those in the stability state (+1SD change in GABA, Fig 3C). A potential explanation is that when the reorganization of frontoparietal connectivity occurs under plasticity (versus stability) state, such reorganization allows optimal synchronization between frontal and parietal regions to support mathematical learning. This can allow effective progression from a stage of skill acquisition that is dominated by executive control and intentional processing subserved by the PFC to more automatic processing subserved by the PPC [20,21,25]. In contrast, increased stability, which signified an advanced skill acquisition stage that should be signaled by less controlled and more automatic processing [77], with increased positive connectivity between the PFC and PPC, might indicate difficulty in moving to automatic processing.

Furthermore, a closer examination of the active dlPFC-tRNS group (blue color in Fig 3) suggests that increased positive connectivity is detrimental both in the stability state (+1SD change in GABA, Fig 3C) and also in the plasticity state (−1SD change in GABA, Fig 3A). These findings tie in with our aforementioned findings that participants with weaker positive baseline frontoparietal connectivity experienced enhanced learning outcomes following dlPFC-tRNS. Therefore, dlPFC-tRNS seem to be beneficial to those with weaker changes in positive baseline frontoparietal connectivity and may be detrimental to those with stronger changes in positive baseline connectivity, regardless of the state (plasticity or stability) (for details see S7 Table). Relatedly, we also report that across changes in GABA concentration (Fig 3B), changes in connectivity predict learning. Specifically, in the sham condition (red panels in Fig 3B), those with a greater increase in positive connectivity exhibited better learning. However, the administration of tRNS reverses this pattern by benefiting those with the least increase in positive frontoparietal connectivity while degrading the performance of those with the greatest change in positive frontoparietal connectivity. At the same time, tRNS did not significantly alter functional connectivity, suggesting that while its effect depends on baseline functional connectivity and its changes, if it causes changes, it is via other mechanisms.

Moreover, our findings expand previous studies that have shown that the effect of brain stimulation varies among individuals [76,41,78–82]. We highlight a key determinant of brain stimulation outcome on neurophysiology, namely, the initial levels of frontoparietal connectivity. Some previous studies in children showed a detrimental influence of stronger functional connectivity in participants with lower mathematical abilities [83,84]. However, the relationship between functional connectivity and math performance is more complex than originally thought and it is influenced by several factors including

developmental stage (children versus adults), concurrent math measures versus learning studies, and in response to interventions (e.g., before versus after math tutoring). For example, greater IPS connectivity with the premotor cortex and supramarginal gyrus has been associated with better performance in adulthood [85]. However, IPS connectivity was negatively associated with math performance in adolescence [86] as in children. Moreover, frontohippocampal connectivity was shown to be negatively correlated with concurrent math measures in adolescence [86], which contrasts with results from learning studies. For example, tutoring-induced strengthened connectivity between IPS with lateral PFC (as well as VTOC, and hippocampus) was also associated with improved mathematical learning in children [87]. In previous studies, we have reported that the association between different brain measures (e.g., GABA, glutamate, aperiodic exponent) and cognition (e.g., math) can be reversed as a function of the developmental stage [15,88]. Future studies should examine whether and how the different brain indices (GABA, glutamate, functional connectivity, aperiodic exponent) are linked to each other and may reflect the changes in the association between functional connectivity and math as a function of the developmental stage. Our results provide important considerations for future individual-specific brain stimulation interventions aimed at enhancing academic learning in typically developing adults or those who suffer from learning difficulties.

### The causal role of dlPFC in calculation but not drill learning in adults

This study provides novel insights into the neural mechanisms underlying mathematical learning, specifically in terms of the role of dlPFC in calculation but not drill learning in adults. Namely, in contrast to previous fMRI studies that highlighted the role of dlPFC also in drill learning, our findings suggest that dlPFC may instead be a correlate and may indicate redundancy in brain functions [89] with regards to drill learning in the adult brain. While we included a mathematical control task, future studies should include a non-mathematical learning control task that also heavily depends on executive functions to help distinguish between specific executive function components. This will allow the examination of whether the observed neurostimulation effect is specific to mathematical learning or due to a domain-general modulation of executive functions. Nevertheless, our findings are theoretically consistent with current meta-analyses and models of arithmetic processing [32,35,90–94]. Our two-time point (i.e., after versus before tRNS) additional analyses did not provide evidence that tRNS modulated working memory/executive functions (S8 Table). Recent tRNS experiments that used a similar prefrontal montage as in this study, improved sustained attention performance or symptoms that are associated with sustained attention impairments [42,49,95]. Therefore, we hypothesize that our results are due to sustained attention enhancement that has been linked with improved academic learning [96]. In this case, the lack of effect on drill learning may be explained by task difficulty. This is in line with previous tRNS studies that have shown that in line with the stochastic resonance framework, task difficulty moderates the effect of tRNS [71,97]. Namely, the presence of a dlPFC-tRNS effect for calculation learning might not necessarily be due to the fact the dlPFC is less involved in drill learning but because drill is an easy task and less demanding compared to calculation. Highly task engagement is thought to boost the effectiveness of non-invasive brain stimulation [98]. Based on the stochastic resonance framework [97] it is expected that the more difficult the task the more the benefit from tRNS. Our findings, which are in line with previous work in the field of cognitive learning [41,71], match these predictions.

### Conclusions

By elucidating the roles of functional connectivity and neurochemicals in learning, our findings contribute to a deeper understanding of how to optimize external interventions for cognitive improvement [99]. These insights hold promise for advancing educational practices, improving socioeconomic outcomes, and promoting better health at both individual and societal levels.

### Supporting information

**S1 Text. Additional Materials and Methods information.**
(DOCX)

**S1 Table. (A–D)** Additional statistical analyses examining the role of baseline dlPFC-hippocampus connectivity and the role of baseline PPC–hippocampus connectivity in predicting academic learning (in the sham tRNS condition). Based on previous work showing that cortico-hippocampal connectivity is involved in math learning [36], we assessed whether dlPFC–hippocampus (or PPC–hippocampus) connectivity would explain variance in calculation or drill. To assess this possibility, we initially examine whether any of these four measures (i.e., dlPFC–hippocampus and PPC–hippocampus*left and right hemisphere) could predict learning using the day, learning type (drill versus calculation), baseline connectivity, and we included random intercepts for participants. **Statistics:** Value=regression coefficient, SE=standard error, DF=degrees of freedom, T=T-value, P=p-value, CI_L=confidence interval lower bound, CI_U=confidence interval upper bound, the suffix "_L" indicates the left hemisphere and the suffix "_R" indicates the right hemisphere.
(DOCX)

**S1 Fig.** [1]H-MRS voxel (2 × 2 × 2 cm) positions for (A) the PPC, and (B) dlPFC are shown on coronal slices.
(DOCX)

**S2 Fig. Plotting the significant two-way interaction of left dlPFC–hippocampus connectivity with learning type from S1 Table D, where more positive connectivity was associated with poorer calculation learning, but no differences were observed for drill learning.** The data underlying the results in S2 Fig can be found in S4 Data.
(DOCX)

**S2 Table. Additional statistical analyses examining the role of baseline frontoparietal connectivity after controlling for baseline dlPFC–hippocampus connectivity in predicting academic learning. Statistics:** Value=regression coefficient, SE=standard error, DF=degrees of freedom, T=t-value, P=p-value, CI_L=confidence interval lower bound, CI_U= confidence interval upper bound, the suffix "_L" indicates the left hemisphere. Interactor predictors are denoted by the * symbol.
(DOCX)

**S3 Fig.** Predicting learning (RT) in the drill learning task using left, Panel **A**, or right, Panel **B** frontoparietal connectivity across three levels of frontoparietal connectivity. Predicting RT in the drill learning task using left, Panel **C** or right, Panel **D** frontoparietal connectivity across three levels of right frontoparietal connectivity, −1SD (left panel), Mean (middle panel), and +1SD (right panel) and as a function of tRNS condition. The data underlying the results in S3 Fig can be found in S2 Data.
(DOCX)

**S3 Table. To establish whether the functional connectivity results we observed are specific to frontoparietal connectivity rather than overall connectivity we assessed whether the control connectivity measures interacted with tRNS condition and type in predicting academic learning.** To this end, we substituted frontoparietal connectivity with any of the four control connectivity measures: (i) left dlPFC–left occipital pole, (ii) right dlPFC–right occipital pole, (iii) left PPC–left occipital pole, and (iv) right PPC–right occipital pole. As can be seen in the tables below none of these four connectivity measures interacted with tRNS conditions and type to predict academic learning as the corresponding three-way interactions were not significant. **Statistics:** Value, regression coefficient; SE, standard error; DF, degrees of freedom, *T*, *T*-value; *P*, *p*-value.
(DOCX)

**S4 Table. A table depicting the statistical results of the linear mixed-effects model predicting learning based on learning type (drill, calculation), tRNS condition (sham tRNS which is the reference group here, dlPFC-tRNS, PPC-tRNS), day (S4 Table A** for reaction time, same as Table 1, and **S4 Table C** for accuracy), and a similar model that additionally featured the three covariates, mathematical attainment, first block calculation reaction time, and

**first block drill reaction time (S4 Table B for reaction time and S4 Table D for accuracy). Statistics:** Value, regression coefficient; SE, standard error, DF, degrees of freedom; $T$, $t$-value; $P$; $p$-value; CI_L, confidence interval lower bound; CI_U, confidence interval upper bound. Interactor predictors are denoted by the * symbol. The displayed "0.0" values in the accuracy sections of the table are due to rounding.
(DOCX)

**S5 Table. dlPFC-tRNS or PPC-tRNS did not significantly alter the levels of GABA and glutamate as shown by the independent sample $t$ test results. Statistics:** $T$, $T$-value; DF, degrees of freedom, $P$, $p$-value; CI_L, confidence interval lower bound; CI_U, confidence interval upper bound.
(DOCX)

**S6 Table. A table depicting the statistical results of the linear mixed-effects model predicting learning based on learning type (drill, calculation), tRNS condition (sham tRNS which is the reference group here, dlPFC-tRNS), dlPFC GABA concentration, day, and right frontoparietal connectivity (denoted as "FC"). Statistics:** Value, regression coefficient; SE, standard error; DF, degrees of freedom; $T$, $t$-value; $P$, $p$-value. Interactor predictors are denoted by the * symbol. **Neurochemicals:** dlPFC, dorsolateral prefrontal cortex; GABA, gamma-aminobutyric acid. A "Δ" prefix denotes a post-tRNS minus pre-tRNS difference score in that neurochemical or functional connectivity measure.
(DOCX)

**S7 Table. Additional analyses that unpack the interplay of Δ frontoparietal connectivity level (denoted as Δ Conn in the table below) and Δ dlPFC GABA on calculation learning separately for the plasticity state (i.e., Plasticity (Δ dlPFC GABA −1SD) and the stability state (i.e., Δ dlPFC GABA +1SD), for the sham tRNS condition (S7 Table A) and the dlPFC-tRNS condition (S7 Table B).** To do this, we extracted the 95% CI from Fig 3 and compared the two subgroups (i.e., Δ frontoparietal connectivity +1SD versus Δ frontoparietal connectivity −1SD). The CI to P value conversion was obtained from the following three steps **(A)** calculate the standard error: $SE = (Upper\ CI − Lower\ CI)/(2 \times 1.96)$ **(B)** calculate the test statistic: $z = Est/SE$ **(C)** calculate the $P$-value: $P = \exp(−0.717 \times z − 0.416 \times z^2)$ and the outcome of each step is documented in the tables below. This formula works only for positive $z$, so if $z$ was negative, the minus sign was removed [1]. **Statistics:** CI_L, confidence interval lower bound; CI_U, confidence interval upper bound.
(DOCX)

**S4 Fig. Plotting the four-way interaction of learning type*tRNS condition*right frontoparietal connectivity across three levels of condition dlPFC GABA, −1SD (Panel A), Mean (Panel B), and +1SD (Panel C), for ease in interpretability here we merely display the drill learning.** The data underlying the results in S4 Fig can be found in S3 Data.
(DOCX)

**S5 Fig. Spectra plots and LCmodel estimates. (A)** The spectra from each region (dlPFC, PPC, and V1) and each time point (Pre, before tRNS; Post, after tRNS) separately. Spectra from all participants are overlaid. **(B)** The LCmodel fit estimates from each region (dlPFC, PPC, and V1) and each neurochemical (GABA=GABA, Glu=glutamate) separately before the tRNS. Fit estimates from all participants are overlaid. **(C)** The LCmodel fit estimates from each region (dlPFC, PPC, and V1) and each neurochemical (GABA=GABA, Glu=glutamate) separately after the tRNS. Fit estimates from all participants are overlaid.
(DOCX)

**S8 Table. Independent sample $t$ test comparing the Sham tRNS vs. dlPFC-tRNS (A) or vs. PPC-tRNS (B) for the following scores: [Attention scores derived by the Attention Network Task (1) where Executive: ANT Executive network, Orienting: ANT Orienting network; Short term memory assessed by the Digit span task (2) and the Corsi Block tapping task (3) tests that included both forward and backward conditions].** A "Δ" prefix denotes a post-tRNS

minus pre-tRNS difference score in each cognitive measure. **Statistics:** *T*, *t*-value, DF, degrees of freedom; *P*, *p*-value; SE, standard error; CI_L, confidence interval lower bound; CI_U, confidence interval upper bound.
(DOCX)

**S9 Table. Additional statistical analyses examining the role of baseline frontoparietal connectivity in predicting accuracy. Statistics:** Value, regression coefficient; SE, standard error; DF, degrees of freedom; *T*, *T*-value; *P*, *p*-value; CI_L, confidence interval lower bound; CI_U, confidence interval upper bound. Interactor predictors are denoted by the * symbol.
(DOCX)

**S10 Table. A table depicting the statistical results of the linear mixed-effects model predicting accuracy based on learning type (drill, calculation), tRNS condition (sham tRNS which is the reference group here, dlPFC-tRNS), dlPFC GABA concentration, day, and right frontoparietal connectivity (denoted as "FC"). Statistics:** Value, regression coefficient; SE, standard error; DF, degrees of freedom; *T*, *T*-value; *P*, *p*-value. Interactor predictors are denoted by the * symbol. **Neurochemicals:** dlPFC, dorsolateral prefrontal cortex; GABA, gamma-aminobutyric acid. A "Δ" prefix denotes a post-tRNS minus pre-tRNS difference score in that neurochemical or functional connectivity measure.
(DOCX)

**S11 Table. Participants' guesses of the tRNS condition at the end of learning.** Cells contain the number of responses of each type. There was a similar response to condition (active tRNS vs. sham tRNS) in each group (Fisher's exact test dlPFC-tRNS vs. sham tRNS, *P* = 0.33, PPC-tRNS vs. sham tRNS, *P* = 0.53).
(DOCX)

**S12 Table. Mean and standard deviation (SD) of reaction time and accuracy separately for learning type (Drill, Calculation), stimulation groups (Sham-tRNS, dlPFC-tRNS, and PPC-tRNS), day (1 to 5), and block.**
(DOCX)

**S13 Table. Minimum Reporting Standards for MRS in MRS checklist.**
(DOCX)

**S1 Data. Datafile containing the data underlying Table 1.**
(CSV)

**S2 Data. Datafile containing the data underlying Fig 2 and S3 Fig.**
(CSV)

**S3 Data. Datafile containing the data underlying Fig 3 and S4 Fig.**
(CSV)

**S4 Data. Datafile containing the data underlying S2 Fig.**
(CSV)

## Author contributions

**Conceptualization:** Roi Cohen Kadosh.

**Data curation:** George Zacharopoulos.

**Formal analysis:** George Zacharopoulos, Roi Cohen Kadosh.

**Funding acquisition:** Roi Cohen Kadosh.

**Investigation:** Beatrix Krause-Sorio.

**Methodology:** George Zacharopoulos, Masoumeh Dehghani, Beatrix Krause-Sorio, Roi Cohen Kadosh.

**Project administration:** George Zacharopoulos, Beatrix Krause-Sorio, Roi Cohen Kadosh.

**Resources:** Masoumeh Dehghani, Jamie Near.

**Software:** George Zacharopoulos, Masoumeh Dehghani, Jamie Near.

**Supervision:** Jamie Near, Roi Cohen Kadosh.

**Validation:** Jamie Near.

**Visualization:** George Zacharopoulos.

**Writing – original draft:** George Zacharopoulos, Roi Cohen Kadosh.

**Writing – review & editing:** George Zacharopoulos, Masoumeh Dehghani, Beatrix Krause-Sorio, Jamie Near, Roi Cohen Kadosh.

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
