## [Editor Report · Decision Letter 0]

Dear Dr Zacharopoulos, 

Thank you for submitting your manuscript entitled "Unveiling Causal Mechanisms of Academic Learning Through Neurostimulation and Multimodal Imaging" for consideration as a Research Article by PLOS Biology.

Your manuscript has now been evaluated by the PLOS Biology editorial staff and I am writing to let you know that we would like to send your submission out for external peer review.

Once your full submission is complete, your paper will undergo a series of checks in preparation for peer review. After your manuscript has passed the checks it will be sent out for review. To provide the metadata for your submission, please Login to Editorial Manager (https://www.editorialmanager.com/pbiology) within two working days, i.e. by Oct 27 2024 11:59PM.

Kind regards,

Christian

Christian Schnell, PhD

Senior Editor

PLOS Biology

cschnell@plos.org

---

## [Decision Letter · Decision Letter 1]

Dear Dr Zacharopoulos,

Thank you for your patience while your manuscript "Unveiling Causal Mechanisms of Academic Learning Through Neurostimulation and Multimodal Imaging" was peer-reviewed at PLOS Biology. It has now been evaluated by the PLOS Biology editors, an Academic Editor with relevant expertise, and by several independent reviewers. 

In light of the reviews, which you will find at the end of this email, we would like to invite you to revise the work to thoroughly address the reviewers' reports.

As you will see below, the reviewers overall think that the study is well executed and provides potentially important insights. However, Reviewer 1 and Reviewer 3 in particular raise a few concerns about the lack of a clear hypothesis and statements that are currently not fully supported by the experimental data. 

Given the extent of revision needed, we cannot make a decision about publication until we have seen the revised manuscript and your response to the reviewers' comments. Your revised manuscript is likely to be sent for further evaluation by all or a subset of the reviewers.

**IMPORTANT - SUBMITTING YOUR REVISION**

*Re-submission Checklist*

*Published Peer Review*

*PLOS Data Policy*

*Blot and Gel Data Policy*

Sincerely,

Christian

Christian Schnell, PhD

Senior Editor

PLOS Biology

cschnell@plos.org

REVIEWS:

Reviewer #1: This study aims to qualify the causal contributions of the DLPFC, PPC, and hippocampus to mathematical learning. tRNS was applied to the DLPFC or PPC, and the results were compared to those with sham tRNS. Among several findings, a particularly interesting result was that participants with poorer baseline learning performance and weaker frontparietal positive connectivity showed enhanced learning following tRNS to the DLPFC.

While the above findings from the multimodal approach are intriguing, I have several concerns:

1. Heuristic Nature of the Manuscript:

The entire manuscript is written in a highly heuristic manner. The interpretations of the results are not grounded in clear hypotheses or predictions derived from established models. Instead, interpretations appear speculative, lacking further experiments or theoretical frameworks to support them. This makes the manuscript overly speculative, reducing its scientific rigor.

2. Lack of Hypotheses and Counter-Hypotheses:

While hypotheses and predictions are mentioned, the manuscript does not explain why these specific hypotheses or predictions were made. Furthermore, no counter-hypotheses are discussed, which would strengthen the interpretation of the results and demonstrate a more balanced approach.

3. Speculative Conclusions:

It feels as though the multimodal approach yielded interesting results, but these findings were then used as a basis for speculation rather than being tied to robust experimental evidence or theoretical models.

Recommendations for Improvement:

1. Elaborate on Hypotheses and Predictions:

Clearly state the hypotheses and predictions, ensuring they are derived from prior experimental results, theoretical models, or logical reasoning. Including counter-hypotheses would also provide a more comprehensive framework for interpreting the results.

2. Conduct Additional Experiments:

Perform further experiments to validate the speculations derived from the current findings. This would lend greater credibility to the interpretations and strengthen the manuscript's overall contribution.

By addressing these points, the manuscript would be in significantly better shape, offering a more robust and scientifically rigorous contribution to the field.

Reviewer #2: I think this is a well thought out and important piece of experimental work. I have very few comments.

I note the often change in MRS measured metabolite levels are described as "neurotransmitter" levels. This is not accurate. Please adjust this language throughout. Please define what exactly being is measured using MRS (cellular/synaptic/extracellular) to give the general reader a clearer understanding. Also please expand upon why MRS measured glutamate and GABA can be taken as a proxy for E/I - at what levels? I'd suggest including with appropriate references the evidence that the MRS substrate quantified may be a reasonable reflection of neurotransmitter levels (there are preclinical studies to this effect) and this can subsequently support discussion of results in terms of a neurosignalling response. 

Still on MRS - could the manuscript please be updated to confirm with MRS reporting guidelines - especially as they have a co-author of this guidance.

Reviewer #3: This interventional study employs a multimodal approach to investigate the influence of electrical stimulation (tRNS) on learning mathematical procedures and encoding arithmetic facts. It examines behavior, functional connectivity, and transmitter chemistry (glutamate and GABA levels). The authors report that individuals with stronger fronto-parietal connectivity prior to training achieved greater learning success in mathematical procedures. Those individuals with poorer fronto-parietal connectivity subsequently benefited from frontal tRN stimulation. Regarding transmitter chemistry, a complex four-way interaction was found.

The multimodal approach of this study is novel, very interesting and promising. The manuscript is well written, the sample size should be adequate, although on the smaller side. The study has the potential to yield valuable academic insights into mathematical network processes. However, in its current form, the manuscript suggests evidence for an applicable (and practically relevant) intervention for individuals with difficulties in learning mathematics, which, in my opinion, cannot be derived from the data presented here for several reasons.

Major Comments:

- The authors find significant effects exclusively for mathematical procedures (e.g., the relationship between stronger connectivity and learning success, or between frontal tRNS and learning success) but not with memory-based drill training. Mathematical procedures in arithmetic heavily rely on EF like working memory, attention processes, inhibition, or shifting (see Cragg & Gilmore, 2014 for a review), typically assumed to be associated with the DLPFC. In line with this, the modulation observed in this study occurred only with tRNS over the DLPFC. However, the study lacks a non-mathematical control task that also heavily depends on EF, let alone control tasks that might help distinguish between specific EF components. As a result, it cannot be concluded that the observed effect is specific to mathematical learning.

Despite this, the authors draw precisely this conclusion (p. 19): "This study provides novel insights into the neural mechanisms underlying mathematical learning." An alternative interpretation would be that the findings represent a domain-general modulation of EF, which does not need to be inherently specific to mathematical learning. This is not necessarily a limitation; in fact, the results might suggest implications for EF trainings, which in turn could indirectly enhance mathematical procedures. Nevertheless, the results should be interpreted and discussed more cautiously.

That said, the authors could emphasize more that their findings are nevertheless theoretically consistent with current meta-analyses and models of numerical processing. According to these models, mathematical procedures and calculation impose high demands on EF, reflected in strong frontal activation within the math-responsive network. Meanwhile, encoding arithmetic facts is supposed to be stronger associated with medial temporal areas, as well as inferior parietal and temporal cortices (e.g., Amalric & Dehaene, 2018; Arsalidou & Taylor, 2011; Arsalidou et al., 2018; Hawes et al., 2019; Klein & Knops, 2023; Menon, 2016; Siemann & Petermann, 2018).

This opens up an alternative explanation for the absence of effects during drill training. Instead of interpreting this as "It might indicate that the effect of tRNS on mathematical drill learning is less robust," it could simply be that other brain regions critical for drill training—such as the hippocampus—were not targeted by tRNS in this study (which, indeed, cannot be stimulated directly).

- There are other conclusions that are not justified in my opinion. The authors conclude (p. 15): "Our results, in line with previous results (6-8), support the view that such differences are primarily of biological, rather than environmental, origin. Importantly, our results highlight an approach that can reverse such disadvantage using an excitatory form of neurostimulation to target the neural substrates involved in calculation learning to abolish the Matthew effect." These claims are problematic for several reasons: 1) The study does not provide evidence to definitively determine whether differences in learning mathematical procedures are of biological rather than environmental origin as this was not tested in the study. To draw such a causal conclusion, environmental factors, such as socioeconomic status (SES), would need not only to be controlled for but also systematically manipulated (e.g., through the inclusion of an additional group). 2) The claim that the study demonstrates the ability of tRNS to "reverse" disadvantages associated with the Matthew effect is also unfounded. On the one hand, this presupposes a long-term effect, which was not tested here and therefore cannot be assumed. The measurements were conducted immediately after the five-day training, and potential long-term effects of electrical stimulation are, at best, highly debated. On the other hand, this presupposes that the improvement effects are within a considerable range that has real practical relevance (ideally not only within the ms range of reaction times, but also relevant accuracy improvements etc.) instead of rather academic relevance (for the elicitation of model-theoretical questions, however, this study is definitely novel and interesting). However, I could not find accuracy data in the supplementary materials (SM), nor data plots on training curves (e.g., fewer repetitions during training could have been an interesting indicator). Even the reaction time data analysis is hidden in Figures S6A and S6B, and these effects do not appear particularly large. Therefore, I would tone down the claims regarding practical relevance. Together with the first major point, the correction of this discussion and conclusions requires, in my opinion, reframing of the manuscript.

- Since „stronger baseline positive connectivity" showed better learning performance in this study, I am missing a discussion with previous data showing detrimental influence of stronger functional connectivity in participants with lower mathematical abilities (e.g., Rosenberg-Lee et al., 2015; Jolles et al., 2016; Abreu-Mendoza et al. 2022; Michels et al., 2018).

- Perhaps the training data could also provide further information regarding the complex 4-fold interaction of GABA levels. Depending on the initial GABA level/functional connectivity, the strongest training improvement might be reached earlier than after 5 training sessions (steepness of the performance improvement during training, e.g. in accuracy data) or has not yet been reached at all in other participants. At the very least, this would be interesting to see in the SM.

- At the beginning of the methods section, it should be made explicit and clear to readers that participants were divided into three groups, and demographic data should be given for these three groups (this information is currently provided in the SM). While the grouping becomes clearer later in the manuscript, at this point, it initially gave the impression of a within-subject design. Of course, given the potential for long-term effects, a within-subject design would not be feasible, but it is important to clarify from the outset that this study examines between-subject effects, along with all the associated limitations.

Minor comments:

- The results of GABAergic modulation are not clearly stated in the abstract.

- Instead of an introduction, the manuscript has a summary, probably a typo

- There should also be references for the Matthew effect in the introduction (p. 3), not just in the discussion. 

- p. 3: „Longitudinal studies indicate that mathematical abilities are relatively stable from childhood through adulthood, driven primarily by biological rather than environmental factors (6-8)." But there are also alternative suggestions, according to Judd et al. (2020; PNAS), this is independently influenced by socioeconomic status and polygenetic factors.

- p. 4: "…mathematical learning by modulating brain activity and neurochemicals involved…". -> "…mathematical learning by cortical excitability and neurochemicals involved… (Nitsche et al., 2005)".

- P. 5: "tRNS, due to its suggested excitatory effect is hypothesized to enhance plasticity later in life". Again, this sounds as if it has been proven and recognized that a permanent change in neuro-cognitive plasticity can be achieved over a lifetime by using tRNS in 5 sessions. If there is a study that has shown this, please cite it here (and elsewhere), otherwise such statements do not belong here.

- P. 8: "A scan was performed…" -> "A scan (i.e., H-MRS and resting state fMRI) was performed…"

- In my opinion, the behavioral data results should be included in the main manuscript, along with a table in the SM containing descriptive statistics of the behavioral data. These are essential for readers to evaluate the magnitude of the manipulations.

- Figure 2 is a bit confusing as in panels B and C the colours indicate the values for mean and SDs, while in D and E they indicate stimulation type. Would it be possible to make the layout of the illustrations more uniform or at least use different colors?

- P. 14: "In this study, we examined how tRNS can alter mathematical learning…" Again, due to the lack of a non-mathematical control task this is hard to defend.

---

## [Editor Report · Decision Letter 2]

Dear Dr Zacharopoulos,

Thank you for your patience while we considered your revised manuscript "Unveiling Causal Mechanisms of Academic Learning Through Neurostimulation and Multimodal Imaging" for publication as a Research Article at PLOS Biology. This revised version of your manuscript has been evaluated by the PLOS Biology editors, the Academic Editor and one of the original reviewers.

Based on the reviews and on our Academic Editor's assessment of your revision, we are likely to accept this manuscript for publication, provided you satisfactorily address the following data and other policy-related requests:

* We would like to suggest a different title to improve its accessibility for our broad audience: "Neurostimulation modulates the interplay between functional connectivity and GABAergic signaling to enhance mathematical learning"

* Please add the links to the funding agencies in the Financial Disclosure statement in the manuscript details.

* Please mention in the "Competing interests" section that Roi Cohen Kadosh is an Editorial Board Member at PLOS Biology.

* Please include the approval/license number of the institutional review board.

* Please include information in the Methods section whether the study has been conducted according to the principles expressed in the Declaration of Helsinki.

* Please specify whether the participants provided written or oral consent.

* DATA POLICY:

Regardless of the method selected, please ensure that you provide the individual numerical values that underlie the summary data displayed in the following figure panels as they are essential for readers to assess your analysis and to reproduce it: 2BCDE, 3, SI2E, SI4ABC and SI10.

* CODE POLICY

We expect to receive your revised manuscript within two weeks. 

*Published Peer Review History*

*Press*

Sincerely,

Christian

Christian Schnell, PhD

Senior Editor

cschnell@plos.org

PLOS Biology

---

## [Editor Report · Decision Letter 3]

Dear Dr Zacharopoulos,

Thank you for your patience while we considered your revised manuscript "Functional connectivity and GABAergic signalling modulate the enhancement effect of neurostimulation on mathematical learning" for publication as a Research Article at PLOS Biology. This revised version of your manuscript has been evaluated by the PLOS Biology editors.

Thank you for addressing the editorial requests, which are mostly completely addressed. However, we need you to upload the source data and reference them in the corresponding figure legends before we can accept the manuscript for publication.

I am pasting the details again below:

* DATA POLICY:

Regardless of the method selected, please ensure that you provide the individual numerical values that underlie the summary data displayed in the following figure panels as they are essential for readers to assess your analysis and to reproduce it: 2BCDE, 3, SI2E, SI4ABC and SI10.

* CODE POLICY

We expect to receive your revised manuscript within two weeks. 

*Published Peer Review History*

*Press*

Sincerely,

Christian

Christian Schnell, PhD

Senior Editor

cschnell@plos.org

PLOS Biology

---

## [Editor Report · Decision Letter 4]

Dear Dr Zacharopoulos,

Thank you for the submission of your revised Research Article "Functional connectivity and GABAergic signalling modulate the enhancement effect of neurostimulation on mathematical learning" for publication in PLOS Biology. On behalf of my colleagues and the Academic Editor, Simon Hanslmayr, I am pleased to say that we can in principle accept your manuscript for publication, provided you address any remaining formatting and reporting issues. These will be detailed in an email you should receive within 2-3 business days from our colleagues in the journal operations team; no action is required from you until then. Please note that we will not be able to formally accept your manuscript and schedule it for publication until you have completed any requested changes.

When you attend to those requests to come, please also provide the source data for the supplementary figures SI2E, SI4ABC and SI10. Please also mention in those figure legends where the source data can be found.

PRESS

We frequently collaborate with press offices. If your institution or institutions have a press office, please notify them about your upcoming paper at this point, to enable them to help maximize its impact. If the press office is planning to promote your findings, we would be grateful if they could coordinate with biologypress@plos.org. If you have previously opted in to the early version process, we ask that you notify us immediately of any press plans so that we may opt out on your behalf.

Sincerely, 

Christian

Christian Schnell, PhD

Senior Editor

PLOS Biology

cschnell@plos.org